# Enzymatic Potential of Filamentous Fungi as a Biological Pretreatment for Acidogenic Fermentation of Coffee Waste

**DOI:** 10.3390/biom12091284

**Published:** 2022-09-12

**Authors:** Joana Pereira, Ana Cachinho, Marcelo M. R. de Melo, Carlos M. Silva, Paulo C. Lemos, Ana M. R. B. Xavier, Luísa S. Serafim

**Affiliations:** 1Department of Chemistry, CICECO-Aveiro Institute of Materials, Universidade de Aveiro, 3810-193 Aveiro, Portugal; 2LAQV-REQUIMTE, Faculty of Science and Technology, Universidade NOVA de Lisboa, 2829-516 Caparica, Portugal

**Keywords:** acidogenic fermentation, enzymatic hydrolysis, *Paecilomyces variotii*, short-chain organic acids, solid-state fermentation, spent coffee grounds, *Trametes versicolor*

## Abstract

Spent coffee grounds (SCGs) are a promising substrate that can be valorized by biotechnological processes, such as for short-chain organic acid (SCOA) production, but their complex structure implies the application of a pretreatment step to increase their biodegradability. Physicochemical pretreatments are widely studied but have multiple drawbacks. An alternative is the application of biological pretreatments that include using fungi *Trametes versicolor* and *Paecilomyces variotii* that naturally can degrade complex substrates such as SCGs. This study intended to compare acidic and basic hydrolysis and supercritical CO_2_ extraction with the application of these fungi. The highest concentration of SCOAs, 2.52 gCOD/L, was achieved after the acidification of SCGs pretreated with acid hydrolysis, but a very similar result, 2.44 gCOD/L, was obtained after submerged fermentation of SCGs by *T. versicolor*. This pretreatment also resulted in the best acidification degree, 48%, a very promising result compared to the 13% obtained with the control, untreated SCGs, highlighting the potential of biological pretreatments.

## 1. Introduction

Coffee is one of the most popular and traded beverages in the world with a steady consumption growth of 1.9% per year in the last ten years [1], leading to an estimated production for 2020/21 of over 10 Gt [2]. During coffee processing, huge amounts of lignocellulosic waste are produced, usually ending up in landfills or burned to generate energy [3]. These strategies could be problematic since the high contents of caffeine and phenolic compounds, particularly in spent coffee grounds (SCGs), negatively affect the environment. On the other hand, SCGs are a residue that contains polysaccharides, proteins, and lipids, and, consequently, are an interesting feedstock for chemical and bio-based processes to obtain high-value products for pharmaceutical, cosmetic, and food industries [4]. In fact, in the Coffee Development Report, released in 2020 by the International Coffee Organization, the valorization of coffee waste through the production of bioproducts and biofuels was highlighted as a key strategy to reduce the environmental impact of coffee production [1].

SCGs can be valorized in a myriad of ways, including through the production of short-chain organic acids (SCOAs). SCOAs are aliphatic monocarboxylic acids with two to six carbon atoms, such as acetic, propionic, butyric, isobutyric, valeric, caproic, and lactic acids, with applications in several industries, either directly or as a building block for further conversion [5]. These molecules are usually produced by petrochemical processes, but the environmental impact and the increasing cost of crude oil promoted the interest in biological production, especially using organic waste as a substrate [6].

Microbial production of SCOAs can use either pure-culture fermentations for a targeted acid or use mixed culture in acidogenic fermentation (AF), which is one of the steps of anaerobic digestion for a mixed stream of acids. Mixed-culture fermentation is particularly suited for waste use and has lower production costs [7]. The obtained mixtures of SCOAs can be used for the production of polyhydroxyalkanoates (PHAs) with mixed cultures [8]. The competitiveness of the AF process needs to be improved to make its implementation feasible. Strategies for enhancing SCOA production should focus on improving hydrolysis rates to produce more soluble substrates for further fermentation, promoting the acidogenic process by manipulating operating parameters, and/or removing inhibitory sources, such as the presence of methanogens or undissociated acids [9].

The focus on hydrolysis is crucial since this is the rate-limiting step due to the complexity of wastes and the presence of recalcitrant compounds. This situation often requires the introduction of a pretreatment step to improve substrate solubility and sugar availability by removing lignin and hemicelluloses, reducing the cellulose crystallinity, and increasing the surface area for posterior biological breakdown [10]. The selection of the ideal pretreatment highly depends on the substrate, and usually, physical and/or chemical technologies are employed. However, due to technical, economical, and environmental drawbacks of physicochemical pretreatments, the development of biological strategies has been rising.

Biological pretreatments often apply microorganisms different from those in AF; fungi are the most common, since they are typically very efficient in hydrolyzing and degrading the most complex substrates [11]. The fungal species chosen for pretreatments bear specific enzymatic systems, and they can be combined considering the feedstock composition. Fungal hydrolytic processes usually have lower energy requirements, raising the cost-effectiveness of the process [11,12]. Furthermore, biological treatments are generally non-polluting since they produce fewer waste streams, or at the very least, they are less harmful [11]. The use of fungi as a pretreatment was already shown as effective when applied to AF. The SCOA production from wheat straw, after delignification by the white-rot fungi *Phanerochaete chrysosporium*, was similar to that obtained after alkaline hydrolysis pretreatment [12]. The pretreatment of wastewater biosolids with an enzymatic mixture of cellulases and xylanases led to an 86% increase in the SCOA yield [13]. The combination of biological and physicochemical pretreatments can also be advantageous, resulting in increases of 97% and 143% in the concentrations of acetic and butyric acids, respectively, after the AF of sweet sorghum stalks pretreated with NaOH and cellulases [14].

Filamentous fungi have been exploited for the industrial production of enzymes for many decades, and extensive studies on suitable fermentation conditions and more efficient strains were found [15]. The use of enzymatic cocktails is the most common approach regarding biological pretreatments. However, the high price of enzyme concentration and purification is a significant production cost. These enzymes can be produced through both submerged fermentation (SmF) and solid-state fermentation (SSF). In the former, microorganisms grow dispersed in the liquid medium, whereas in the latter, they grow directly on the substrate with a limited moisture content [16]. Both strategies have advantages and disadvantages, and the most suitable process is highly dependent on the substrate and strain chosen. SmF is reported to have better results due to higher oxygen availability [17] and mixing and is advantageous regarding instrumentation and control, biomass separation, and scaling-up. On the other hand, SSF mimics the natural growing conditions of fungi, is less susceptible to substrate inhibition, and has lower demands of energy and operating costs [16,18].

Considering the chemical structure of SCGs, two species of filamentous fungi, *Trametes versicolor* and *Paecilomyces variotii,* were selected to be tested as biological agents for the pretreatment step in the present work.

*T. versicolor*, as a white-rot fungus, plays an important ecological function as a primary decomposer of wood through the production of ligninolytic enzymes, especially lignin peroxidase (LiP), manganese peroxidase (MnP), cellobiose dehydrogenase, and laccase isoenzymes (Lac) [18]. Due to their wide degrading spectrum and intense oxidative action, these enzymes degrade complex products other than lignin, such as xenobiotic compounds, pesticides, aromatic hydrocarbons, and chlorinated organic compounds, among others [19]. Besides ligninolytic enzymes, *T. versicolor* is also reported to have a high cellulolytic activity [20].

Similarly, *P. variotii* can be found worldwide in soils, animals, indoor environments, and food products. Its ability to grow in common agro-industrial derivatives and degrade their toxic contaminants, aliphatic and aromatic hydrocarbons, is a consequence of the numerous enzymes produced, such as amylases, chitinases, pectinases, phytases, tannases, and xylanases [21].

The inclusion of a pretreatment step before the AF of SCGs was already proven to be beneficial using different types of physical and chemical technologies, namely, acid and basic pretreatments and supercritical CO_2_ extraction [22]. In the present study, besides testing the SCG pretreatment with non-purified enzymatic extracts produced by *T. versicolor* and *P. variotii*, the use of SmF and SSF as a pretreatment were also assessed. Finally, to understand the benefits and impact of the biological pretreatment, the results were compared to physicochemical pretreatments already reported as efficient for the AF of SCGs.

## 2. Materials and Methods

### 2.1. Substrate

SCGs were collected at the coffee shop of the Department of Chemistry of the University of Aveiro and dried in an oven at 105 °C to a constant weight for 24 h, and then stored in a desiccator.

### 2.2. Microorganisms

*Paecilomyces variotii* NRRL-115 was supplied by the Agricultural Research Service Culture Collection at the National Center for Agricultural Utilization Research, USDA; *Trametes versicolor* CBS 109428 was supplied by the Westerdijk Fungal Biodiversity Institute. Cultures were maintained in Petri dishes with malt extract agar (30 g of malt extract, 5 g of peptone, and 15 g of agar per liter of distilled water) and kept at 4 °C. Every month the fungi were replicated to ensure a fresh and viable culture.

### 2.3. Enzyme Production

For inoculum preparation, the mycelium was removed from the Petri dishes and suspended in *Trametes* Defined Medium (TDM), prepared as described in reference [23]. Then, 1 mL of suspension was used to inoculate 250 mL of TDM with glucose in 500 mL Erlenmeyer flasks and incubated at 28 °C and 180 rpm. Three days after inoculum preparation, the culture was filtered, washed with TDM, and transferred to 500 mL Erlenmeyer flasks with 250 mL of TDM without glucose and incubated at 28 °C and 180 rpm for 14 days. At the end of the fermentation assay, the remaining medium was centrifugated at 5000 rpm for 1 h (Centrifuge Thermo, Thermo Fisher, Waltham, MA, USA). The pellet was discarded, and the supernatant was filtered with a paper filter (0.45 μm) under sterile conditions and stored at −16 °C.

### 2.4. SCG Pretreatments

#### 2.4.1. Physicochemical

The physicochemical pretreatments chosen to be performed with 1.5 g of SCGs suspended in 40 mL of deionized water were as follows: acid hydrolysis with 5% H_2_SO_4_ for 1 h in the autoclave at 121 °C; basic hydrolysis with 2% of NaOH in the autoclave at 121 °C for 1 h; and supercritical CO_2_ extraction at 300 bar and 50 °C, with a flow rate of 12 gCO_2_/min for 2 h (0.5 L lab unit, Speed-SFE model from Applied Separations, Inc, Allentown, PA, USA), as reported by Marcelo et al. [24]. The pretreatment results were applied isolated or in combination in the acidogenic fermentation assays, as summarized in Table 1.

#### 2.4.2. Biological

Enzymatic hydrolysis was performed using 1.5 g of SCGs in 40 mL of the enzymatic extracts obtained in Section 2.3. at 40 °C and 100 rpm for 7 d. SmF was conducted by inoculating the selected fungus in 500 mL Erlenmeyer flasks with TDM without glucose and incubating at 28 °C and 180 rpm for 14 d. At the end of the assay, the remaining medium was centrifugated at 5000 rpm for 1 h (Centrifuge Thermo, Thermo Fisher). The SCG pellet was sterilized under UV radiation for 30 min, dried at 105 °C for 24 h, and stored to be further used in AF assays. SSF was performed in cotton-plugged 250 mL Erlenmeyer flasks using 5 g of SCGs as substrate with a moisture content of 1:4 (*w*/*v*). The substrate was inoculated with a 3 cm-diameter agar plug from a fully grown plate and the cultures were incubated at 28 °C for 3 months. An overview of the experimental conditions is presented in Table 1.

### 2.5. Acidogenic Fermentation

#### 2.5.1. Inoculum

The inoculum was collected from an aerobic tank of SIMRia, the wastewater treatment plant (WWTP) in Aveiro Sul, Aveiro (Portugal), and maintained at 4 °C.

#### 2.5.2. Experimental Set-Up

To study the pretreatment effect on the AF of SCGs, batch tests were conducted in flasks with 100 mL of working volume. To each flask, 0.15 g of chemical oxygen demand (COD) of aerobic sludge and 1 g of COD of SCGs previously submitted to the different pretreatments (Table 1) were added and supplemented with a mineral solution (160 mg/L of NH_4_Cl, 160 mg/L of KH_2_PO_4_, 80 mg/L of CaCl_2_, 160 mg/L of MgSO_4_, 800 mg/L of NaHCO_3_, 200 mg/L of CoCl_2_, 30 mg/L of MnCl_2_, 10 mg/L of CuCl_2_, 100 mg/L of ZnSO_4_, 300 mg/L of H_3_BO_3_, 30 mg/L (NH_4_)_6_Mo_7_O_2_, and 20 mg/L of NiCl_2_). The flasks were encapsulated and purged with N_2_, to ensure anaerobic conditions, and incubated at 28 °C and 300 rpm. Every 24 h, a 2.0 mL sample was collected under anaerobic conditions and centrifuged at 13,000 rpm for 10 min (MiniSpin, Eppendorf, Hamburg, Germany). The pellet was discharged, and the pH of the supernatant was assessed. The supernatant was stored at −16 °C for further determination of SCOA, glucose, and xylose concentrations.

### 2.6. Analytical Methods

#### 2.6.1. Determination of SCOAs and Monomeric Sugars

For each sample, 600 μL was filtered using VectaSpin Tubes (Whatman, Piscataway, NJ, USA) with a membrane of 0.2 μm (Whatman, Kent, UK) at 8000 rpm (MiniSpin Eppendorf, Hamburg, Germany) for 20 min before HPLC injection. Monomeric sugar concentration of the pretreatment extracts was determined using a Rezex RPM-Monosaccharide Pb^+2^ (8%) column (Phenomenex, Torrance, CA, USA) at 85 °C, with a refractive index detector (Merck, Darmstadt, Germany) that used MilliQ water as an eluent (0.6 mL/min). SCOA concentration of the acidogenic tests was measured in a Rezex ROA-Organic Acid H^+^ (8%) column (Phenomenex, Torrance, CA, USA) at 65 °C, alongside a refractive index detector (Merck, Darmstadt, Germany) that used H_2_SO_4_ 0.005 N as an eluent (0.5 mL/min). The calibration curves were carried out using synthetic compounds (Sigma-Aldrich, St. Louis, MO, USA) in the range of 0–1 g/L for sugars and 0–5 g/L for SCOAs.

#### 2.6.2. Chemical Oxygen Demand (COD)

COD was measured with the Spectroquant Kit (Merck Millipore, Darmstadt, Germany), and the solutions used were prepared according to standard methods [25]: a digestive aqueous solution made with K_2_Cr_2_O_7_, HgSO_4_, and H_2_SO_4_ and an acid solution made with H_2_SO_4_ and AgSO_4_. To 2 mL of the properly diluted sample, 1.2 mL of digestive solution and 2.8 mL of acid solution were added. The mixture was incubated at 150 °C for 2 h. After cooling, the absorbance was measured. The calibration was performed with glucose with COD concentrations between 0–1 g/L.

### 2.7. Calculations

#### 2.7.1. COD Conversions

For further calculations, the concentrations of SCOAs determined by HPLC were converted from g/L to gCOD/L using conversion factors that represent the mass (g) of oxygen required to oxidize 1 g of a compound based on the oxidation reactions for each compound. The overall oxidation equation is represented by:a compound + b O_2_ → c CO_2_ + d H_2_O + e NH_3_(1)
where a, b, c, d, and e represent the stoichiometric coefficients of the equation. Therefore, the conversion factor (cf) was calculated according to the following equation:(2)$$\mathrm{cf}\;\left({\mathrm{gO}}_{2}/g\right)=\frac{b\;\times \;\mathrm{MM}\left(O_{2}\right)}{a\;\times \;\mathrm{MM}\left(\mathrm{compound}\right)}$$
where MM corresponds to the molar mass. The conversion factors were 1.07 gO_2_/g for glucose, xylose, and acetate; 1.51 gO_2_/g for propionate; 1.82 gO_2_/g for butyrate; 2.04 gO_2_/g for valerate.

#### 2.7.2. Acidification Degree (AD)

The COD values were used for the calculation of the acidification degree (AD), which represents the amount of substrate consumed to produce SCOAs by considering the organic matter feed in the batch assays, and was calculated using Equation (3). These calculations were performed as percentages.
(3)$$\mathrm{AD}\left(\mathrm{gCOD}/\mathrm{gCOD}\right)=\frac{\mathrm{SCOAproduced}}{\mathrm{CODin}}\times 100$$

#### 2.7.3. Yields, Rates, and Productivities

Sugar yield was calculated by dividing the total amount of monomeric sugars in the pretreatment extract by the amount of SCGs multiplied by 100. COD yield was calculated by dividing the total COD extracted in the pretreatment by the COD of the SCGs. The initial rate of production for each acid was calculated by adjusting a linear function to the experimental data of SCOA concentrations plotted over the initial days of operation and calculating the first derivative at time zero (taking the slope of the fitting). SCOA volumetric productivity was calculated by dividing the amount of produced SCOAs in grams of COD by volume and time.

#### 2.7.4. Odd-to-Even Ratio of SCOAs

With further valorization of SCOAs into PHAs in mind, the odd-to-even ratio of acids was calculated to evaluate the potential for each monomer production, according to [22]. It was defined as the sum of odd-equivalent carboxylic acids formed (propionic and n-valeric acids) divided by the sum of even-equivalent carboxylic acids formed (acetic and n-butyric acids), according to Equation (4).
(4)$$\mathrm{Odd-to-Even}\;\mathrm{Ratio}=\frac{\left[\mathrm{Propionic}\right]\;+\;[\mathrm{n-Valeric}]}{\left[\mathrm{Acetic}\right]\;+\;[\mathrm{n-Butyric}]}$$

## 3. Results and Discussion

### 3.1. Pretreatment Efficiency

The ideal pretreatment should allow high sugar recovery while limiting the presence of inhibitor compounds [26]. Since extreme pretreatment conditions can result in the formation of microbial inhibitors such as furans and phenolic compounds, a posterior detoxification step could be necessary, increasing the process cost [27,28]. For this reason, in the present work, several pretreatments with mild conditions were applied to SCGs. The release of monomeric sugars and the extraction yield for the physicochemical pretreatments are described in Table 2. No monomeric sugars were detected in the biological pretreatments.

Acid hydrolysis was the pretreatment that resulted in the highest concentration of monomeric sugars, both when applied to normal SCGs (AH)—2.23 g/L, corresponding to a yield of 5.95 gSugar/gSCG, and to SCGs submitted to supercritical CO_2_ extraction (SC+AH)—1.51 g/L, corresponding to a yield of 4.03 gSugar/gSCG (Table 2). This pretreatment is often reported as one of the most effective for sugar release from SCGs [29,30]. The highest value obtained in this study is in range of what was obtained by other authors: using the acid hydrolysis under similar conditions, a concentration of 1.95 g/L was achieved by Pereira et al. [22], and working with SCGs submitted to a microwave pretreatment and 1.5% H_2_SO_4_, Lopéz-Linares et al. [29] reported an extraction of 5.8 g/L of sugars [29]. However, the sugar yields obtained were low compared with some works that reported values in the range of 20 to 50% [31,32]. It is important to notice that, although the concentrations of acid used were similar, the temperatures applied were higher (above 160 °C) and/or the reaction times were longer (3 h). Thus, parameters such as acid concentration, hydrolysis time, and temperature should be optimized for SCGs to increase AH efficiency.

Pretreatment with NaOH (BH) led to sugar concentrations near zero, 0.03 g/L, corresponding to a yield of 0.08 gSugar/gSCG when performed alone and in combination with SC. This was expected since alkaline agents, when combined with a thermal pretreatment, promote delignification by swelling fibers and loosening macromolecules [33]. Still, some authors reported better sugar yields with this process on SCGs using higher concentrations of NaOH and combined with other pretreatments [34,35]. This might not be the ideal strategy as harsher conditions would also extract high amounts of inhibitory compounds, such as phenolics and furans, that have been proven to have a detrimental impact on AF [22].

Supercritical extraction (SC) was already used to extract oil and diterpenes from SCGs [24,36], and as such, an aqueous phase with solubilized released sugars could not be obtained. However, it was expected that the removal of the lipidic fraction could cause structural changes to the SCG matrix, increasing the accessibility for chemical hydrolysis [26]. In this work, the concentration of sugars obtained for the SCGs submitted to SC was 32% lower and like those obtained with non-pretreated SCGs for the acid and the basic pretreatments, respectively. Some researchers reported similar extraction rates regardless of the oil extraction methodology [31], whereas others reported a 20.8% increase in total sugars when submitting defatted SCGs to AH [37]. These studies used organic solvents to extract the oil fraction of the SCGs by Soxhlet and not supercritical extraction, which might explain the differences observed. This could indicate that the oil extraction method could play a crucial role in the process and the lower sugar yields in SC+AH could be linked to partial degradation of hemicelluloses during supercritical extraction [38].

Regarding the results of enzymatic hydrolysis, despite being often applied to enhance monosaccharide production [26], in this work, as expected, no monomeric sugars were observed in the hydrolysate obtained in any of the biological pretreatments conducted [39,40]. The enzymatic extracts produced by *P. variotii* and *T. versicolor* are usually rich in oxidative enzymes that act on lignin and phenolic compounds [41]. The enzymatic extracts obtained in the current work were tested for Lac, LiP, and MnP, confirming the oxidative nature. It was expected that the application of the fungal extracts to SCGs would facilitate the subsequent acidogenic process due to lignin degradation and degradation of toxic compounds [41,42].

### 3.2. Acidogenic Fermentation

#### 3.2.1. Physicochemical Pretreatments

The results of the AF using SCGs submitted to physicochemical pretreatments are detailed in Figure 1 along with the results for the control (non-pretreated SCGs). The maximum concentration of SCOAs achieved, as well as the initial production rate, are summarized in Table 3. In all assays, a diverse profile of SCOAs was obtained, and all pretreatments resulted in concentrations higher than the control. In these assays, the pH was initially adjusted to 6.0 and then was left uncontrolled. As expected, the pH dropped in most assays due to the production of SCOAs. The decrease was lower for the assays with BH due to the alkali nature of the pretreatment, which probably produces some compounds that could act as buffering agents [22,43].

The control assay (Figure 1A) demonstrated that the aerobic inoculum could acidify the SCGs and produce a combination of acetic, propionic, butyric, and valeric acids with a proportion of 47.5/32.5/15.0/5.0% for the day with the highest production of SCOAs. The highest concentration of SCOAs, 1.31 gCOD/L, was achieved on day 26. This value was higher than the value previously obtained, 0.52 gCOD/L [22], and probably resulted from the higher substrate/biomass ratio used in this work. This result was very similar to what was observed by Arroja et al. [44] (1.33 gCOD/L) after optimization of the process on a moving bed biofilm reactor (MBBR) [45].

As occurred for the hydrolysis, SCG_AH (Figure 1C) was the most efficient process delivering the highest SCOA concentration, 2.52 gCOD/L. The production of acetic, propionic, butyric, and valeric acids on this assay corresponded to a proportion of 35.6/16.9/45.7/1.8% for the day with the maximum production. In the first days, lactic acid was the predominant SCOA produced. This could result from the low pH, as pH values under 6 were reported to promote lactic acid production [46]. Given that similar pH values were also observed in the other assays with no lactic acid production, this was probably a consequence of the microorganisms’ adaptation to the substrate, possibly due to the release of toxic compounds [44]. As AH was the most efficient pretreatment in terms of sugars extraction, it may also have led to the formation of furans and phenolic compounds that the culture had to adapt to. Still, after day 6, a shift in SCOA production was observed, and lactic acid decreased to residual values. The maximum SCOA production was achieved on day 28, and it corresponded to a 93% increase compared to the control.

The assay SCG_AH stood out due to the high production of butyric acid compared to the other SCOAs, with a maximum on day 18 of 1.15 gCOD/L. A clear preference for the production of even acids (acetic and butyric) was already observed in a previous work with the AF of SCOAs treated with AH [22], and several other authors reported the dominance of even acids when working with other types of feedstocks. Zhang et al. [9] reached similar SCOA concentrations when pretreating corn stover with diluted nitric acid. Acetic and propionic acids were the dominant SCOAs during the first 48 h, but then butyric acid was substituted for propionic acid [9]. Using brewery spent grain pretreated with thermal diluted hydrolysis (1.5% H_2_SO_4_), Castilla-Archilla et al. [46] tested AF without a pH control and with a pH controlled at 6.0 [46]. In both assays, butyric acid was the dominant SCOA, and acidic conditions seemed to favor butyric acid production while acetic acid increased with the increase in pH. Kumar et al. [47] also reported similar results when combining an autoclave with 1% H_2_SO_4_ to pretreat vegetal waste [47].

Although basic hydrolysis (SCG_BH, Figure 1E) has proven to result in terms of lignin degradation [10], its effect as a pretreatment for the AF of SCGs is still not clear. This work led to the second-best concentration of SCOAs out of the single physicochemical pretreatments, 2.21 gCOD/L, corresponding to an increase of 69%. This represented an improvement compared to a previous study where the alkali pretreatment, with 5% NaOH, underperformed compared to other strategies, presumably due to the formation of inhibitory compounds during hydrolysis [22]. In this case, lowering the concentration of NaOH to 2% seemed to be an effective solution to prevent this effect and increase SCOA concentration. However, Girotto et al. [48] reported the opposite effect when testing different NaOH concentrations (2, 4, 6, and 8% *w*/*w*) to pretreat SCGs for 24 h at room temperature since a direct relationship between the increase in NaOH concentration and SCG lignin degradation was observed [48]. In another study, the highest concentration of NaOH, 8% (*w*/*w*), resulted in a maximum SCOA production approximately 17 times higher than the one obtained in the present work [49]. The differences observed could result from the different temperatures and hydrolysis times applied, 24 h at room temperature, as they can have a synergistic effect on hydrolysis with NaOH [50]. Furthermore, other authors have reported improvements in the AF of other types of substrates submitted to low concentrations of NaOH (<1%) [12].

Regarding the SCOA profile for the day of highest production, SCG_BH led to a proportion of 7.0/37.8/48.3/3.1/3.8% of lactic, acetic, propionic, isobutyric and valeric acids, respectively. Additionally, working with SCGs, Girotto et al. [49] obtained mainly acetic acid (77%), as well as lower concentrations of propionic, butyric, and valeric acids. Using other waste sludges with the alkali pretreatment for AF, other authors reported similar concentrations of SCOAs but higher concentrations of isobutyric, butyric, and valeric acids [51,52]. However, the type of substrate and MMC acclimatization have a more significant impact on the SCOA profile than pH, as valeric acid and other complex SCOAs often require the bioconversion from macromolecular proteins, and thus, are highly influenced by the protein content of the substrate.

SC as a pretreatment for SCGs led to very remarkable results, both as an individual pretreatment and in combination with chemical hydrolysis. The highest initial production rates were achieved for this pretreatment (Table 3), which is consistent with the effect of SC extraction on SCGs [24]. By removing the oil from the SCGs, the surface area was increased and the microbiological access to the substrate facilitated; therefore, SCOA production occurs faster [26,36]. In the assay with SCGs submitted only to SC (SCG_SC, Figure 1B), a significant improvement (38%) can be observed in comparison to the control, with SCOA production reaching a maximum of 1.80 gCOD/L on day 22. In a previous study, this increase was not observed, and the SC-pretreated SCGs led to slighter lower SCOA concentrations than the non-pretreated waste [22], which might be due to different compositions of the MMC or longer acidification times used in this work. Authors working with other wastes submitted to lipid extraction also reported increases in SCOA production, suggesting that the pretreatment was capable of degrading part of the proteins and carbohydrates into amino acids and smaller sugar chains, thus increasing biomass digestibility for further processing [53,54]. The higher degradation of proteins could explain the profile of SCOAs obtained: 47.3/21.9/3.6/21.6/5.5% of acetic, propionic, isobutyric, butyric, and valeric acids, respectively. The proportion of valeric acid obtained was the highest of all the physicochemical pretreatments tested, and the presence of this acid is often related to the presence of proteins [11].

When SC-pretreated SCGs were submitted to AH and BH, the results were similar in terms of evolution to the ones obtained for non-treated SCGs (Figure 1D,C and Figure 1F,E for AH and BH, respectively), but the maximum concentrations reached were slightly lower than the ones obtained without the SC pretreatment. Considering SCOA profiles, similar proportions of acids were observed for the assays with the alkali treatment, whereas for the acidic treatment, the use of SCGs submitted to SC led to a higher production of acetic acid and no production of butyric acid. Despite not increasing the production of acids, the combination of SC and hydrolysis is still advantageous for the process when considering a biorefining perspective. In this way, SCG oil can be removed and valorized without a significant impact on SCOA production.

#### 3.2.2. Biological Pretreatments

The possibility of replacing physicochemical pretreatments with a more sustainable and cheaper biological alternative by applying enzymatic extracts (EH) of *T. versicolor* and *P. variotii* or directed submerged (SmF) and solid-state (SSF) fermentations with both fungi was tested with SCGs prior to their AF. The obtained results are shown in Figure 2 and the production metrics detailed in Table 2.

SCOAs were produced in all assays, but in most cases, the concentrations reached were lower than the ones obtained in the control assay (SCG_C, Figure 1A).

The use of SCG_TvSmF as a pretreatment positively affected SCOA production, reaching a maximum of 2.44 gCOD/L, which was an 87% increase related to the control. Acetic, propionic, and butyric acids were produced in an average proportion of 59.9/33.8/6.3%. Acetic production occurred throughout the assay, and butyric and propionic acids appeared after the 9th and 18th day, respectively. SmF was already tested as a pretreatment for AF. Tsafrakidou et al. [12] used *Trichoderma viride* combined with alkali hydrolysis in wheat straw. Although these authors observed a higher release of glucose and cellobiose, the increase in SCOA production was irrelevant when compared with non-treated wheat straw [12]. This might be linked to the release of other compounds during the alkali hydrolysis step that could have hindered the fungal metabolism, such as furans or phenolic compounds resulting from the degradation of lignocellulosic biomass and sugars, respectively. These could have hindered the fungal metabolism since they are known microbial inhibitors, impacting cell growth, the sugar uptake rate, and DNA, plasmid, RNA, and/or protein synthesis [22,28,55].

On the other hand, when treating SCGs with SCG_PvSmF (Figure 2B), the production of SCOAs (0.89 gCOD/L) was lower than the control, and only acetic (61.3%) and propionic (38.7%) acids were detected. This might indicate that some of the hemicelluloses and celluloses of SCGs were probably degraded and consumed, as was observed before in other studies [56]. Contrary to other assays, acetic acid started to be consumed after 16 days and was exhausted by day 23, with propionic acid the only one left. A similar trend was also observed with EH using the enzymatic extract from the same fungi. It seemed that *P. variotii* spores survived the UV sterilization, contaminated the enzymatic extract, and were able to grow in the fermentation medium and consume acetic acid, as *P. variotii* is a known acetic acid consumer [57]. Although its toleration is reported, propionic acid is not the favored substrate source of this fungus [58]. Nonetheless, this could be an interesting route to further explore to manipulate the SCOA profile during AF.

It was previously proposed that, by mimicking natural conditions of fungal growth, SSF could lead to higher enzymatic production than SmF, and thus, result in a more efficient pretreatment [59]. Furthermore, by promoting fungal delignification the occurrence of lignin degradation products was believed to be lower and the possible inhibition of AF could be avoided [12]. However, for SSF, only when using SCGs pretreated by *T. versicolor*, SCG_TvSSF (Figure 2C) did it lead to a significant SCOA production, whereas SCG_PvSSF (Figure 2D) resulted in less than half of the SCOA production of the control. SCG_TvSSF led to the production of the most varied profile of SCOAs of all the biological pretreatments, with a proportion of 5.2/39.7/29.0/4.5/15.3/6.4% of lactic, acetic, propionic isobutyric, butyric, and valeric acids, respectively, on the day with the highest concentration of SCOAs. The maximum production reached 1.51 gCOD/L on day 25, a 15% increase compared with non-pretreated SCGs. These results show that the use of the pretreatment could be advantageous to the process, but the results obtained were lower than what was reported with the same fungi and other biomass sources. Tišma et al. [60] submitted corn silage pretreated by *T. versicolor* to AF and obtained a maximum production of 7.12 gCOD/L, almost 5 times higher than in this work [60]. Similarly, using wheat straw submitted to SSF by *P. chrysosporium*, Tsafrakidou et al. [12] reached 5.01 and 6.93 gCOD/L of SCOAs with free and immobilized cells, respectively [12]. The discrepancy in results is probably related to the nature of the substrates since SCGs are more complex and have higher concentrations of proteins and lipids [11,61]. Another possible explanation could be associated with the SSF incubation time; in those works, fungi were only incubated for 1 to 3 weeks, whereas in this work, the SSF pretreatment was performed for 3 months. During this time, besides lignin oxidation, *T. versicolor* could have consumed the hemicellulosic fraction of SCGs, lowering the sugars available for AF. In fact, Fang et al. observed higher rates of cellulose degradation in combination with lignin breakdown after a solid digestate pretreated by *T. versicolor* yielded 9 % less SCOAs than the control test [62]. Other studies also reported cellulose degradation when treating the biomass with *T. versicolor*, leading to significant decreases in SCOA production [63,64]. This could justify the low SCOA production of PvSSF, since *P. variotii* is known to have high cellulosic and hemicellulosic activities [65].

Finally, the result of the enzymatic hydrolysis with the extracts produced by *T. versicolor* and *P. variotii* were submitted to AF, as shown in Figure 2E and Figure 2F, respectively. For SCG_TvEH, only acetic (77.9%) and propionic acids were produced, whereas with SCG_PvEH a small proportion of butyric acid was also observed (9.0%). The SCOA evolution was like that obtained in the SmF assays, but the concentrations were lower, with maximum values of 1.23 gCOD/L and 0.97 gCOD/L for SCG_TvEH and SCG_PvEH, respectively. This represents a decrease of 6 and 25% compared to the control. Better results were expected as, although never tested on SCGs, enzymatic hydrolysis is often used and reported to have a positive impact on the AF of other substrates. Rusli et al. [66] observed an increase in SCOA production when pretreating oil palm fronds with enzymatic extracts produced in-house by the white-rot fungus, *Ganoderma lucidum*. After the pretreatment, hemicellulose and lignin contents decreased while cellulose remained unaltered. They suggested that Lac, LiP, and MnP present in the extracts promoted in vitro rumen digestibility and boosted acid production [67]. Other authors also reported improvements in AF when pretreating corn silage [68] and corn bran [66] with Lac. The higher concentrations of SCOAs obtained in these works probably resulted from higher enzymatic activities of the extracts produced, since these were concentrated before being applied to the substrate. It is also possible that the strategy used to remove fungi cells was not the most effective, as mycelium growth was observed during the AF assays, probably due to fungal spores that remained on the treated SCGs. Therefore, the easily accessible SCOAs could have been used for fungi metabolism. In the future, a different strategy to recover active enzymes should be considered to avoid fungal growth during AF, and enzymatic concentration should be evaluated to understand if the costs of this extra step translate into a significant increase in SCOA production. Additionally, other fungus species should be evaluated, specifically those that usually thrive in food wastes, even if they constitute mixed-species populations of unknown composition.

#### 3.2.3. Overall Performance

To understand and compare the overall performance of the different studied assays, the acidification degree (AD) and the volumetric productivity of each one was calculated and presented in Table 4, together with the maximum SCOAs produced and the corresponding odd-to-even carbon ratio. As discussed before, AH led to the highest concentration of SCOAs, followed by TvSmF. All physicochemical pretreatments improved the amount of SCOAs produced in relation to the control, whereas for the biological pretreatments, only TvSmF and TvSSF were able to do it. Still, other factors must be considered when evaluating the best pretreatment strategy for AF.

The odd-to-even ratio of SCOAs can give insightful information about the potential applications of the acidified stream produced. It is a particularly relevant parameter if the end goal is to produce PHAs since its composition can be manipulated by the type of acid fed to the bacteria. Even-number carbon sources will lead to the production of the homopolymer of hydroxybutyrate, whereas odd-number SCOAs result in a copolymer with hydroxybutyrate and hydroxyvalerate [69]. Most assays led to ratios between 0.30 and 0.80; however, a few interesting changes were observed. The highest ratios were obtained for the assays with the alkali pretreatment due to a high prevalence of propionic acid. These results contradict what was found in the literature for SCGs with this type of pretreatment, with reports of ratios of 0.03 [22] and approximately 0.11 [49]. This could be due to the higher concentrations of NaOH used in those studies that resulted in more inhibitors and limited the diversity of the MMC. On the other end, acid treatments resulted in the lowest odd-to-even carbon ratios, with values in the range of what was previously reported (10–55%) [22,70]. The biological pretreatments seemed to have a less clear impact on the odd-to-even ratios as values vary without a clear tendency.

The AD ranged from 5 to 48%, indicating that a significant part of the substrate was not degraded, maybe due to the high recalcitrance of SCGs [70]. The highest AD was obtained with SCG_TvSmF, although this was not the assay with the highest concentration of SCOAs, indicating that this could be a very promising pretreatment for SCGs. This value was just slightly lower than the one obtained by Arroja et al. [45] after optimization on an MBBR reactor. Furthermore, AD is highly impacted by operating parameters, with authors reporting significant improvements by controlling the pH [46] and by manipulating the organic loading rate and the retention times [44,71]; therefore, an optimization of AF to increase AD should be conducted.

Another factor that is dependent on the optimization of the operating parameters is productivity. Values ranged from 0.024 to 0.117 gCOD/L.d, similar to what was obtained in reference [22]. Basic hydrolysis resulted in the highest productivities both with and without SC treatment due to the fast rate of SCOA production. This was also observed before, even though the SCOA concentration was one of the lowest of all the conditions tested, SCG_BH resulted in high productivities. Such results could indicate that the alkali treatment might be more efficient in leaving the substrate readily available for microbial conversion. SCG_TvSmF and acid hydrolysis pretreatments led to the following best results, in combination with the high concentrations of SCOAs obtained. Another interesting aspect is that the control resulted in the second lowest productivity, meaning that even if some pretreatments did not improve the SCOA concentration, they made the process faster.

The selection of the ideal pretreatment must consider all these factors and is dependent on the goal of the process. A balance must be achieved between high rates of production and an efficient treatment of the waste. Additionally, the selection of the ideal pretreatment depends on the scaling-up of the AF process since the use of a continuous process is desirable, especially when the resulting SCOA stream will be used, after the removal of any biomass, in another biological process such as PHA production by MMC.

## 4. Conclusions

This work proved that the inclusion of a pretreatment step can help to make SCGs a suitable substrate for valorization. In general, physicochemical pretreatments yielded better results, with acid hydrolysis delivering the highest SCOA concentration (2.52 gCOD/L). Even though some of the biological pretreatments had worse performances than the control, this strategy still showed a lot of potential, particularly using *T. versicolor*. This fungus seemed the best suited for the process, achieving the second best SCOA concentration (2.44 gCOD/L) and the highest AD (48%) when the pretreatment was conducted in SmF. Unlike physicochemical pretreatments, biological pretreatments were never applied to SCGs for AF; therefore, there is much room for optimization of the process conditions. It would also be interesting to test the use of a mixture of microorganisms, as interactions between different fungi and bacteria could promote a more efficient degradation of the substrate.

Despite being preliminary, the results of this work are a contribution to lowering the costs of using enzymatic hydrolysis as a pretreatment of complex feedstock. The application of the unpurified enzymatic extracts or the fungus directly without any pretreatments can represent a cheaper alternative to the use of purified enzymes.

## Figures and Tables

**Figure 1 biomolecules-12-01284-f001:**
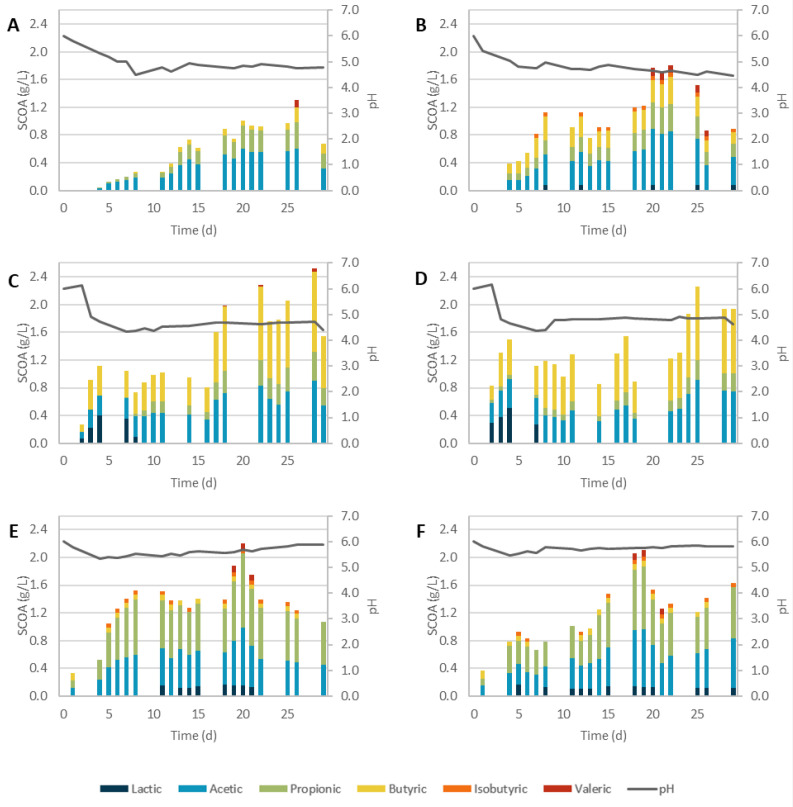
SCOA production during AF of SCGs submitted to physicochemical pretreatments ((**A**): control; (**B**): supercritical CO_2_ extraction; (**C**): acid hydrolysis; (**D**): acid hydrolysis and supercritical CO_2_ extraction; (**E**): basic hydrolysis; (**F**): basic hydrolysis and supercritical CO_2_ extraction).

**Figure 2 biomolecules-12-01284-f002:**
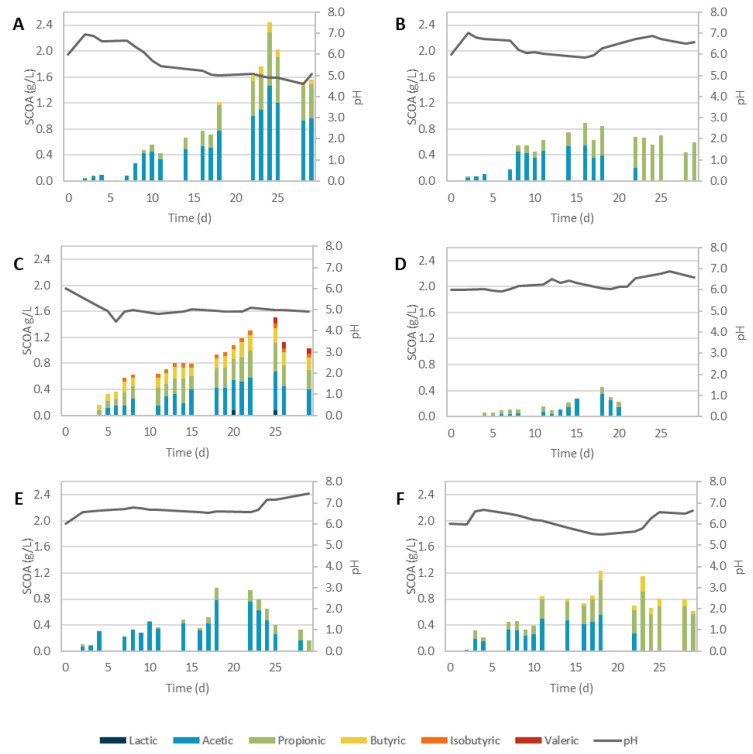
SCOA production during AF of SCGs submitted to biological pretreatments ((**A**): submerged fermentation by *T. versicolor*; (**B**): submerged fermentation by *P. variotii*; (**C**): solid-state fermentation by *T. versicolor*; (**D**): solid-state fermentation by *P. variotii*; (**E**): enzymatic hydrolysis by *T. versicolor*; (**F**): enzymatic hydrolysis by *P. variotii*).

**Table 1 biomolecules-12-01284-t001:** Overview of the pretreatments and respective conditions.

Pretreatment		Experimental Conditions
Acid Hydrolysis	AH	5% H_2_SO_4_ at 121 °C for 1 h in autoclave
Basic Hydrolysis	BH	2% NaOH at 121 °C for 1 h in autoclave
Supercritical Extraction	SC	CO_2_ extraction at 300 bar, 50 °C, 12 gCO_2_/min for 2 h
Supercritical Extraction + Acid Hydrolysis	SC + AH	CO_2_ extraction at 300 bar, 50 °C, 12 gCO_2_/min for 2 h followed by 5% H_2_SO_4_ at 121 °C for 1 h in autoclave
Supercritical Extraction + Basic Hydrolysis	SC + BH	CO_2_ extraction at 300 bar, 50 °C, 12 gCO_2_/min for 2 h followed by 2% NaOH at 121 °C for 1 h in autoclave
Solid-State Fermentation	TvSSF	28 °C for 3 months, 1:4 (*w*/*v*) MC with *T. versicolor*
	PvSSF	28 °C for 3 months, 1:4 (*w*/*v*) MC with *P. variotii*
Submerged Fermentation	TvSmF	28 °C, 180 rpm for 15 d with *T. versicolor*
	PvSmF	28 °C, 180 rpm for 15 d with *P. variotii*
Enzymatic Hydrolysis	TvEH	Enzymatic extract obtained from *T. versicolor* at 40 °C, 100 rpm for 7 d
	PvEH	Enzymatic extract obtained from *P. variotii* at 40 °C, 100 rpm for 7 d

TvSSF—solid-state fermentation by *T. versicolor*; PvSSF—solid-state fermentation by *P. variotii*; TvSmF—submerged fermentation by *T. versicolor*; PvSmF—submerged fermentation by *P. variotii*; TvEH—enzymatic hydrolysis using the enzymatic extract obtained from *T. versicolor*; PvEH—enzymatic hydrolysis using the enzymatic extract obtained from *P. variotii*.

**Table 2 biomolecules-12-01284-t002:** Monomeric sugar concentration and extraction yield for each pretreatment.

Pretreatment	Sugars (g/L)	%Yield (gSugar/gSCG)
AH	2.23	5.95
BH	0.03	0.08
SC + AH	1.51	4.03
SC + BH	0.03	0.08

**Table 3 biomolecules-12-01284-t003:** Production metrics for the acidogenic assays using SCGs submitted to the different pretreatments.

Assay	Lactic		Acetic		Propionic		Isobutyric		Butyric		Valeric	
	SCOAs _Max_ †	r _initial_ *	SCOAs _Max_ †	r _initial_ *	SCOAs _Max_ †	r _initial_ *	SCOAs _Max_ †	r _initial_ *	SCOAs _Max_ †	r _initial_ *	SCOAs _Max_ †	r _initial_ *
SCG_C	-	-	0.56	0.025	0.39	0.007	-	-	0.21	0.036	0.10	0.020
SCG_AH	0.40	0.097	0.90	0.081	0.43	0.024	-	-	1.15	0.132	0.05	0.003
SCG_BH	0.17	0.041	0.83	0.080	1.07	0.089	0.07	0.014	0.08	0.037	0.09	0.042
SCG_SC	0.08	0.005	0.85	0.058	0.40	0.028	0.06	0.030	0.39	0.048	0.11	0.051
SCG_AH + SC	0.50	0.124	0.92	0.108	0.28	0.022	-	-	1.06	0.155	-	-
SCG_BH + SC	0.15	0.031	0.84	0.078	0.90	0.097	0.06	0.011	0.18	0.042	0.10	0.032
SCG_TvSSF	0.08	0.026	0.60	0.054	0.44	0.030	0.07	0.027	0.24	0.025	0.10	0.032
SCG_PvSSF	-	-	0.34	0.022	0.11	0.016	-	-	-	-	-	-
SCG_TvSmF	-	-	1.46	0.025	0.54	0.059	-	-	0.15	0.018	-	-
SCG_PvSmF	-	-	0.55	0.047	0.87	0.020	-	-	-	-	-	-
SCG_PvEH	-	-	0.56	0.050	0.91	0.015	-	-	0.24	0.009	-	-
SCG_TvEH	-	-	0.78	0.066	0.19	0.015	-	-	-	-	-	-

* gCOD/L; † gCOD/L.d.; - means that the compound was not detected.

**Table 4 biomolecules-12-01284-t004:** Overview of the assays’ performances (maximum SCOA concentration and its odd-to-even carbon ratio, acidification degree, and productivity).

Assay	pH _Max SCOA_	SCOAs (gCOD/L)	Odd-to-Even Ratio	AD (%)	Prod. (gCOD/L.d)
SCG_C	4.82	1.31	0.60	13%	0.050
SCG_AH	4.71	2.52	0.23	23%	0.090
SCG_BH	5.70	2.21	1.28	22%	0.111
SCG_SC	4.65	1.8	0.38	31%	0.082
SCG_SE + AH	4.86	2.26	0.14	12%	0.090
SCG_SE + BH	5.75	2.1	1.01	14%	0.117
SCG_TvSSF	5.00	1.51	0.60	15%	0.060
SCG_PvSSF	6.03	0.45	0.31	5%	0.024
SCG_TvSmF	4.90	2.44	0.51	48%	0.102
SCG_PvSmF	5.85	0.89	0.63	12%	0.056
SCG_TvEH	6.61	0.97	0.25	19%	0.054
SCG_PvEH	5.52	1.23	0.77	26%	0.068

## Data Availability

Not applicable.

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
