# Peer review of "Enzymatic Potential of Filamentous Fungi as a Biological Pretreatment for Acidogenic Fermentation of Coffee Waste"

_biomolecules, 2022, doi:10.3390/biom12091284_

Round 1

Reviewer 1 Report

Joana Pereira and colleagues evaluated the enzymatic potential of Trametes versicolor and Paecilomyces variotii filamentous fungi as a pre-treatment for acidogenic fermentation of spent coffee grounds. The use of fungi was also compared to physico-chemical pre-treatments, such as acid or basic hydrolyses, supercritical extraction, or combined methods.

The manuscript has been prepared in a concise and clear way. The introduction is sufficient, and the methodology was appropriately described. The results were clearly presented in tables and figures, as well as, were discussed with other works.

I had only one suggestion regarding the description of the results:

Subsection 3.1. – I propose to add a figure/table for the results of pre-treatment efficiency.

Author Response

The manuscript makes a great effort on evaluating the influence of different pretreatment steps on SCG valorization by biotechnological processes. The manuscript is well organized, and most hypotheses are strongly explained. The results comparison between this work and other studies can help the audience to better understand the merits and drawbacks of this research. However, it is not ready to be published without all the following comments/issues being successfully addressed.

The authors thank Reviewer 1 for the kind evaluation of the work and the comments that contributed to improve this article. All changes were made directly in the document, highlighted by the "track change" tool.

  1. As the author mentioned that “the type of substrate and MMC acclimatization have a significant impact on SCOA profile”, what is the protein content (or range) of the SCG collected at the coffee shop? Will different SCG has different protein content which may result in various SCOA production?

The protein content of the substrate can influence the SCOA profile with some authors reporting the formation of higher concentrations of valeric acid when using protein-based substrates, as mentioned in section 3.2.1. and reference [11]. Unfortunately, the protein content of the SCG used in the study was not determined due to time and material constrictions, so that link could not be made.

  1. In lines 399 - 401, what is the reason that “SC pretreated SCG led to slighter lower SCOA concentrations than the non-pretreated waste in the previous study”?

A possible explanation for the differences observed could be related with a different composition of the microbial population or to the longer acidification time used. These hypotheses were added to that section of the manuscript.

  1. Author mentioned several times that “the release of inhibitors during the alkali hydrolysis step could hinder the fungal metabolism”. What is the mechanism of this process?

Previous studies (references 22 and 30) already showed that alkali hydrolysis can lead to the formation of microbial inhibitors as well as a more extensive explanation of the type of inhibitors and their origin. To make the text clearer to the reader those references were added to the text when this effect is mentioned to provide more support to the discussion, besides reference 57 (a review that discusses the impact of inhibitors from lignocellulosic biomass on microbial communities).

Reviewer 2 Report

The manuscript makes a great effort on evaluating the influence of different pretreatment steps on SCG valorization by biotechnological processes. The manuscript is well organized, and most hypotheses are strongly explained. The results comparison between this work and other studies can help the audience to better understand the merits and drawbacks of this research. However, it is not ready to be published without all the following comments/issues being successfully addressed. 

1. As the author mentioned that “the type of substrate and MMC acclimatization have a significant impact on SCOA profile”, what is the protein content (or range) of the SCG collected at the coffee shop? Will different SCG has different protein content which may result in various SCOA production?

2. In lines 399 - 401, what is the reason that “SC pretreated SCG led to slighter lower SCOA concentrations than the non-pretreated waste in the previous study”?

3. Author mentioned several times that “the release of inhibitors during the alkali hydrolysis step could hinder the fungal metabolism”. What is the mechanism of this process?

Author Response

Joana Pereira and colleagues evaluated the enzymatic potential of Trametes versicolor and Paecilomyces variotii filamentous fungi as a pre-treatment for acidogenic fermentation of spent coffee grounds. The use of fungi was also compared to physico-chemical pre-treatments, such as acid or basic hydrolyses, supercritical extraction, or combined methods. The manuscript has been prepared in a concise and clear way. The introduction is sufficient, and the methodology was appropriately described. The results were clearly presented in tables and figures, as well as, were discussed with other works.

The authors thank Reviewer 2 for the kind evaluation of the work and the comments that contributed to improve this article. All changes were made directly in the document, highlighted by the "track change" tool.

I had only one suggestion regarding the description of the results:

Subsection 3.1. – I propose to add a figure/table for the results of pre-treatment efficiency.

Table 2 was added as suggested.